



# Representing effects of surface heterogeneity in a multi-plume eddy diffusivity mass flux boundary layer parameterization

Nathan Arnold[1]

[1]Goddard Space Flight Center, 8800 Greenbelt Road, Greenbelt, Maryland, USA

**Correspondence:** Nathan Arnold (nathan.arnold@nasa.gov)

**Abstract.** Earth System Models (ESMs) typically represent surface heterogeneity on scales smaller than the atmospheric grid, while land-atmosphere coupling is based on grid mean values. Here we present a general approach allowing subgrid surface heterogeneity to influence the updraft thermodynamic properties in a multi-plume mass flux parameterization. The approach is demonstrated in single column experiments with an Eddy Diffusivity Mass Flux (EDMF) boundary layer scheme. Instead of triggering based on grid mean surface values, updrafts are explicitly assigned to individual surface tiles with positive buoyancy flux. Joint distributions of near-surface vertical velocities and thermodynamic variables are defined over individual surface tiles, and updraft properties are drawn from the positive tails of the distributions. The approach allows updraft properties to covary with surface heterogeneity, and updrafts from different tiles maintain distinct properties to heights of several hundred metres. Mass flux contributions to subgrid variances are increased near the surface, but impacts on mean state variables are relatively small. We suggest that larger impacts might be obtained by adding a specialized plume to represent the effects of secondary circulations.

## 1 Introduction

The Earth's surface varies in temperature, wetness, roughness and other characteristics that impact the exchange of heat, moisture and momentum with the atmosphere. This surface heterogeneity has been shown to impact atmospheric boundary layer (ABL) dynamics (Bou-Zeid et al., 2020), the development of clouds (Xiao et al., 2018; Fast et al., 2019; Chen et al., 2020) and precipitation (Shrestha et al., 2014; Gao et al., 2021). Impacts depend on the scale and organization of surface heterogeneity relative to the atmospheric area and processes of interest (Shen and Leclerc, 1995; Avissar and Schmidt, 1998; Poll et al., 2021).

Despite its importance, representation of heterogeneity in surface-atmosphere coupling remains rudimentary in most contemporary Earth system models (ESMs). Such models typically employ atmospheric grid spacing of 10-100 km, while surface elements are represented on smaller scales with a mosaic of tiles or subgrid patches. In most cases, the atmospheric model component uses only a grid mean representation of the surface fluxes, based on aggregation either of fluxes or relevant parameters across subgrid surface elements. Similarly, although flux calculations may be performed at the tile or patch level, they typically employ only grid mean values from the atmosphere (Giorgi and Avissar, 1997).





Neglecting heterogeneity in surface-atmosphere coupling has been shown to produce simulation biases (Manrique-Suñén et al., 2013; Clark et al., 2015), and approaches have been developed to represent heterogeous effects in ABL parameterizations. Molod et al. (2003) pioneered the "extended mosaic" technique, in which a mosaic of surface tiles was effectively extended through the ABL by performing the boundary layer calculations in tile space. Molod et al. (2004) showed that such an approach has significant impacts in a global climate simulation. De Vrese et al. (2016) extended this further with the Vertical

Tile Extension (VERTEX) scheme, which explicitly represents horizontal mixing (blending) between mosaic elements in the atmosphere. More recently, Huang et al. (2022) explored the impact of surface heterogeneity in the Cloud Layers Unified by Binormals (CLUBB; e.g., Bogenschutz et al., 2012) scheme. By including inter-tile variability in the CLUBB surface boundary conditions, they found increased boundary layer temperature and humidity variances as well as modest increases in cloud cover.

In this study, we similarly incorporate heterogeneity into an ABL scheme by modifying the lower boundary conditions. Suselj et al. (2021) implemented an Eddy Diffusivity Mass Flux (EDMF) boundary layer scheme in the NASA GEOS model, which represents the subgrid vertical transport by coherent boundary layer updrafts with a multi-plume mass flux scheme. The original eddy diffusivity component from Lock et al. (2000) was recently replaced with the turbulent kinetic energy (TKE)-based Simplified Higher Order Closure (SHOC) scheme of Bogenschutz and Krueger (2013). Unlike CLUBB, SHOC does not

include prognostic temperature and humidity variances, which limits the ability of a surface boundary condition to propagate upward through the ABL. However, the multi-plume mass flux scheme offers an alternative mechanism by which to propagate surface heterogeneity.

We describe a simple approach to distribute individual EDMF updrafts across surface tiles, allowing tile-level fluxes and states to determine initial updraft properties. This "Distributed" Mass Flux (DMF) approach involves modifying the updraft

lower boundary conditions, analogous to the modifications made by Huang et al. (2022) in CLUBB. The paper is structured as follows. The host model, baseline EDMF parameterization and the DMF modifications are described in Section 2. The experiment design is described in Section 3. Section 4 presents the results, with discussion and conclusions in Section 5.

## 2 Host model and parameterization description

### 2.1 The GEOS model

The heterogeneity parameterization was implemented in version 11.2.0 of the NASA GEOS model (GMAO, 2023). The GEOS model is used for a range of applications, including numerical weather prediction (NWP), production of reanalyses (Gelaro et al., 2017), and seasonal prediction (Molod et al., 2020). Atmospheric horizontal grid spacing ranges from 12 km to 0.5 degrees in the NWP and seasonal applications, respectively. As the effects of heterogeneity are expected to be more significant at larger scales, the 0.5 degree seasonal application is targeted in this study.

The GEOS land surface is partitioned into a mosaic of tiles representing hydrologic catchments defined by local topography (Koster et al., 2000). The Catchment land surface model computes energy and water fluxes across several vertically stacked soil layers and the land-atmosphere interface. Variability on sub-tile scales is also represented in the form of three hydrolog-





ical regimes whose fractional areas are based on topography and tile conditions at a given timestep: (i) a saturated regime corresponding to soil near rivers and streams, (ii) an uphill subsaturated regime, and (iii) a wilting regime, if conditions are

dry enough. Land-atmosphere fluxes of heat and moisture are calculated at this sub-tile level before being aggregated to tile space and ultimately to the atmospheric grid. Surface runoff is calculated from rainwater reaching the saturated fractional surface. In the present study we use only the tile-level aggregated properties, and any sub-tile variability is ignored, although our heterogeneity scheme could be extended to sub-tile scales in future work.

The GEOS atmosphere component employs the Grell-Freitas deep convection scheme (Freitas et al., 2020), and cloud micro-

and macro-physics use an updated form of Bacmeister et al. (2006). Longwave and shortwave radiation is calculated with the Rapid Radiative Transfer Model for GCMs (Iacono et al., 2008). Here we use a development configuration of the boundary layer with the Eddy Diffusivity Mass Flux (EDMF) scheme of Suselj et al. (2021), using diffusivity from the Simplified Higher Order Closure (SHOC) scheme of Bogenschutz and Krueger (2013), described in more detail below.

Finally, the precipitation disaggregation scheme of Arnold et al. (2023) is also employed, which acts to stochastically dis-

tribute atmospheric precipitation across surface tiles. Arnold et al. found that precipitation disaggregation increased the inter-tile standard deviation of surface fluxes by approximately 20%.

## 2.2   The baseline EDMF scheme

In this section we provide a brief description of the baseline EDMF scheme in GEOS, hereafter referred to as the control approach. Further details can be found in Suselj et al. (2021). The EDMF approach is based on a conceptual decomposition

of the subgrid area into fractions associated with coherent organized updrafts and an environment of smaller scale turbulence. The subgrid vertical flux of a model variable $\phi$ is then given by the area-weighted sum of the environmental contribution, represented with eddy diffusivity, and updraft transport based on a mass flux (MF) approach,

$$\overline{w'\phi'} = -a_e K_\phi \frac{\partial \overline{\phi}}{\partial z} + \sum_{n=1}^{N} a_n M_{u,n}(\phi_n - \overline{\phi}), \tag{1}$$

where $a_e$ is the environmental area fraction, $K_\phi$ is the diffusion coefficient, $M_{u,n}$ and $a_n$ are the mass flux and fractional

area of the $n$th updraft, and $\phi_n$ and $\overline{\phi}$ are the updraft and grid mean values of the model prognostic variable. EDMF in GEOS is a multi-plume scheme employing $N$ updrafts. The number of updrafts has varied among recent studies, with $N = 10$ (Suselj et al., 2021), 100 (Witte et al., 2022), and 40 (Chinita et al., 2023). In the present study we set $N = 30$ by default, and examine sensitivity to $N$ in Section 4.4.

The individual updraft mass flux and properties $\phi_n$ are found with a separate plume model. The vertical evolution of updraft

properties $\phi_n$ is governed by

$$\frac{\partial \phi_n}{\partial z} = \epsilon_n(\overline{\phi} - \phi_n), \tag{2}$$



where $\epsilon_n$ is a fractional rate of lateral entrainment. Entrainment is treated as a stochastic process (Romps and Kuang, 2010; Sušelj et al., 2013), in which discrete entrainment events follow a Poisson distribution, varying with height and among the updrafts.

The updraft vertical velocity is found via the steady state equation,

$$\frac{1}{2}\frac{\partial w_n^2}{\partial z} = a_w g\left(\frac{\theta_{v,n}}{\bar{\theta}} - 1\right) - b_w \epsilon_{h,n} w_n^2, \tag{3}$$

where $a_w = 1$ and $b_w = 1.5$ are constants. The first term on the right-hand side represents buoyant acceleration, and the second term incorporates sub-plume variability and pressure perturbation effects (De Roode et al., 2012).

Surface boundary conditions for Eqs. (2) and (3) are found by assuming the updrafts emerge from the positive tail of a normal distribution of vertical velocity in the surface layer, between limits $w_{min} = 1.3\sigma_w$ and $w_{max} = 3\sigma_w$. The standard deviation of vertical velocity, $\sigma_w$, is related to the Deardorff convective velocity $w^*$ (e.g., Stull, 1988),

$$\sigma_w = \alpha_w w^*,$$

where $\alpha_w = 0.286$ is a constant. The distribution tail between $w_{min}$ and $w_{max}$ is divided into $N$ equidistant bins, with the
near-surface updraft vertical velocities $w_n|_s$ equal to the central values from each bin. The updraft thermodynamic properties are parameterized following Cheinet (2003). Building on the work of Mahrt and Paumier (1984), the near-surface vertical velocity, virtual potential temperature, $\theta_v$, and total water mixing ratio, $q_t$, are assumed to follow a joint normal distribution and are positively correlated. Taking the updraft velocities as defined above, the near-surface updraft thermodynamic properties are given by

$$\theta_{v,n}|_s = \overline{\theta_v}|_s + c(w, \theta_v) w_n|_s \frac{\sigma_{\theta_v}}{\sigma_w} \tag{4}$$

$$q_{t,n}|_s = \overline{q_t}|_s + c(w, q_t) w_n|_s \frac{\sigma_{\theta_v}}{\sigma_w} \tag{5}$$

where the correlations $c(w, q_t) = 0.32$ and $c(w, \theta_v) = 0.58$, and the standard deviations are based on the surface sensible heat and moisture fluxes, $\sigma_{\theta_v} = \alpha_\theta \overline{w'\theta_v'}|_s / w^*$, and $\sigma_{q_t} = \alpha_{q_t} \overline{w'q_t'}|_s / w^*$, with $\alpha_\theta$ and $\alpha_{q_t}$ both set to 2.89.

## 2.3  The eddy diffusivity scheme


The eddy diffusivities $K_\phi$ appearing in Eq. (1) are calculated using the Simplified Higher Order Closure (SHOC) scheme (Bogenschutz and Krueger, 2013). They are related to a prognostic TKE, $\bar{e}$, by

$$K_H = \tau_v \bar{e}$$

where $K_H$ is the diffusivity for heat and other scalars, and $\tau_v$ is a damping timescale defined as

$$\tau = \frac{2\epsilon}{\bar{e}}$$





$$\tau_v = \frac{\tau}{1 + \lambda_0 N^2 \tau^2}$$

where $\epsilon$ is the rate of TKE dissipation, $N^2$ is the moist Brunt-Vaisala frequency, and $\lambda_0$ is a constant set to 0.04 when $N^2 > 0$ and zero otherwise.

The TKE evolves according to

$$\frac{\partial \overline{e}}{\partial t} = \frac{g}{\theta_v} \overline{w'\theta'_v} - \overline{w'u'} \frac{\partial \overline{u}}{\partial z} - \overline{w'v'} \frac{\partial \overline{v}}{\partial z} - \frac{\partial \overline{w'e}}{\partial z} - \overline{\boldsymbol{v}} \cdot \nabla \overline{e} - \epsilon$$

with right hand side terms representing buoyancy and shear production, parameterized and resolved TKE transport, and dissipation. The buoyancy flux is calculated from an assumed trivariate analytic double gaussian (ADG) joint probability distribution of vertical velocity, liquid water static energy, and total water specific humidity. The ADG PDF is constrained by higher order moments (variances, covariances and triple products) of the three variables estimated within the SHOC scheme. Further details can be found in Bogenschutz and Krueger (2013), but we note here an important difference in the GEOS implementation, namely that the higher order moments each include a contribution diagnosed from the mass flux. The fluxes follow the EDMF decomposition of Eq. (1), while the variances and covariances follow

$$\overline{\phi'_i \phi'_j} = \tau_v K_\phi \frac{\partial \phi_i}{\partial z} \frac{\partial \phi_j}{\partial z} + \sum_{n=1}^{N} a_{u,n} (\phi_{i,n} - \overline{\phi_i})(\phi_{j,n} - \overline{\phi_j}).$$

In the context of this study, including a mass flux contribution allows surface heterogeneity to directly impact the estimated higher order moments, and through them, the buoyancy flux and TKE. Cloud fraction and condensate are also diagnosed from the ADG PDF. Due to the included mass flux contributions, the ADG PDF in this implementation implicitly represents the entire subgrid area, including the updrafts. We therefore use the PDF to represent the entire cloud field, including shallow cumulus associated with the mass flux, rather than including a separate cloud contribution diagnosed directly from the updrafts.

## 2.4 The Distributed Mass Flux approach

In this section we describe our approach to incorporate surface heterogeneity into the multi-plume EDMF scheme. Conceptually, we distribute the mass flux across the subgrid surface, by assigning a portion of the $N$ updrafts to each tile with a positive surface buoyancy flux and calculating updraft lower boundary conditions based on the individual tile properties.

We first sort the surface tiles by their buoyancy flux, and set aside any tiles where the flux is negative. The $N$ updrafts are divided evenly across the buoyant tiles, and any remainder is added singly to the most buoyant tiles in descending order. If the number of buoyant tiles exceeds the number of updrafts, then a single updraft is assigned to the first $N$ tiles in descending order of their buoyancy flux.

Over each buoyant tile, we assume that the near-surface vertical velocity distribution can be parameterized with a separate instance of the normal distribution described in Section 2.2, now with $w^*$ computed from the local tile buoyancy flux. As in the control scheme, updraft vertical velocities are drawn from the positive tail from $1.3\sigma_w$ to $3\sigma_w$, segmented for the number of updrafts assigned to the tile. The thermodynamic properties are similarly drawn from a joint distribution locally defined over each tile, with the tile-level fluxes of sensible heat and moisture replacing the aggregated fluxes used in Eqs. (4) and (5).





Finally, to represent inter-tile atmospheric variability we include an additional thermodynamic perturbation proportional to the deviation of the tile value from the mean surface value. For example, if $\Delta\theta_{s,i} = \theta_{s,i} - \overline{\theta_s}$ is the surface temperature anomaly of tile $i$, then the lower boundary temperature of updraft $n$ over tile $i$ is given by

$$\theta_{v,n,i}|_s = \overline{\theta_v}|_s + c(w,\theta_v)w_n|_s \frac{\sigma_{\theta_v,i}}{\sigma_{w,i}} + \beta\Delta\theta_{s,i} \tag{6}$$

where $\beta$ is a factor of proportionality between the tile-scale atmospheric anomaly and the surface anomaly, and the $\sigma$ values are defined using the local tile fluxes. The $\beta$ parameter reflects the strength of land-atmosphere coupling, and should ideally depend on a number of factors including stability, wind speed, and the scale of surface heterogeneity. Here we simply set $\beta$ to a default value of 0.25, and examine sensitivity of our results to $\beta$ in Section 4.4. We note that the tile surface properties used in the $\beta$ term are the same used in the bulk formula surface flux computations. The $\beta$ term is analogous to the inter-patch

variance incorporated by Huang et al. (2022) into the CLUBB scheme.

The approach is illustrated in Fig. 1, which depicts the assumed near-surface distributions of a thermodynamic property $\phi \subset \{\theta_v, q_t\}$ over three representative surface tiles. The distribution means are offset from the grid mean $\overline{\phi}$ by the $\beta$ terms, and the distribution widths depend on the tile surface fluxes. Updraft properties are drawn from the shaded segments, with updraft fractional areas proportional to the area under each segment. The intended outcome is that updraft properties will vary with

both the intensity of surface fluxes over a given tile, and the near-surface inter-tile variability. This also allows the updrafts to naturally propagate the surface covariance of $\theta$ and $q_t$ into the boundary layer, in contrast to the control scheme where such covariance is assumed positive, regardless of the surface heterogeneity.

The $\beta$ term introduces the possibility that updrafts over a cold tile could be initialized with a negative buoyancy. To prevent this, a check is added to ensure that updrafts assigned to the tile will remain buoyant at the second model level:

$$\beta\Delta\theta_{s,i} + c(w,\theta_v)w_1|_s \frac{\sigma_{\theta_v,i}}{\sigma_{w,i}} > 0.2(\theta_v^{k_s+1} - \theta_v^{k_s}) \tag{7}$$

If a tile fails this condition, its updrafts are redistributed across the remaining buoyant tiles as described above. Note that this issue is somewhat artificial, arising in part because the updraft buoyancy is evaluated against the atmospheric grid mean $\overline{\theta_v}$ and neglects any subgrid variability. In nature, such updrafts would have positive initial buoyancy relative to the local "tile" area, which should persist for some distance while the updraft approaches the atmospheric blending height. This criterion is

approximate; a more precise estimate could be obtained using the updraft entrainment rate, but this varies stochastically and is determined after the code in question.

## 3 Experiments

The heterogeneous DMF scheme is compared with the control approach in a series of experiments with the GEOS single column model (SCM). The GEOS SCM is simply a runtime configuration of the full GCM executable, in which a simplified

large-scale forcing is used in place of the dynamical core. In this study, the boundary conditions and large scale forcing are based on the ARM Southern Great Plains (SGP) location, from 1 June to 31 August, 2017. The model domain is defined as a half





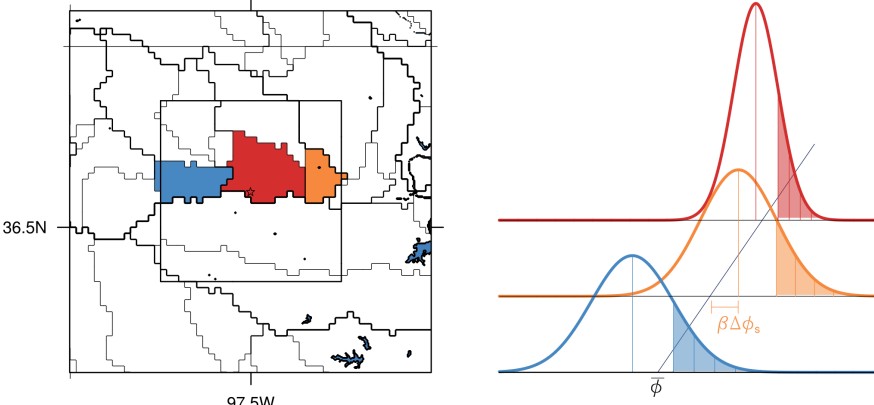

**Figure 1.** (left) Catchment tile boundaries around the ARM SGP site, with 0.5 degree SCM domain (inner box). (right) Illustration of assumed subtile near-surface distributions of generic thermodynamic variable $\phi$. The distributions over each tile are derived from tile-level surface quantities, and updraft properties are drawn from the shaded tails.

degree gridbox centered on the ARM SGP site. Atmospheric initial conditions and advective forcing tendencies are taken from the Variational Analysis continuous forcing dataset (VARANAL; Tang et al., 2019). VARANAL uses a constrained variational analysis to estimate profiles of advective tendencies and state variables based on soundings taken within the SGP domain. To

minimize climate drift over the three month period, the SCM temperature and humidity are relaxed to the VARANAL analyzed profiles with a 48 hour timescale, and relaxation tendencies are further scaled with a height-dependent factor, $(1 + \tanh((z - 500)/250))/2$, to reduce their influence near the surface. At a height of 100 m, this results in an effective relaxation timescale of approximately 50 days. The tile boundaries and model domain are depicted in Fig. 1a. Of those tiles (or partial tiles) within the model grid box, 10 have fractional areas larger than 0.01 and are included in our SCM experiments. We use 137 levels with

vertical grid spacing set to roughly 5 hPa below 700 hPa, then increasing linearly to the model top.

Two configurations of the surface are specified. First, a realistic case in which the tile characteristics are identical to those 10 tiles in the global GEOS model within the SGP domain. Due to the relative homogeneity of the SGP region, all 10 tiles are coded as grassland with similar characteristics, and we label this case "Hom." Second, an enhanced heterogeneity case in which four grassland tiles are replaced with broadleaf deciduous trees based on nearby tiles southeast of the model grid

box, and a fifth grassland tile is replaced with a lake based on the nearby Eufaula Lake in Oklahoma. In subsequent sections, we label this the "Het" case. Surface tile initial conditions for both cases were taken from a global simulation: both land and atmosphere were initialized from MERRA-2, and then run for a further three week period while the atmosphere was constrained by MERRA-2 reanalysis using a "replay" approach (Orbe et al., 2017; Takacs et al., 2018). This is intended to allow some dynamical adjustment by the current model physics while maintaining the reanalysis constraint. Surface trends in both SCM

cases were found to be negligible over the first two weeks, suggesting that the spin up procedure was adequate. Each of these surface configurations was then run with both the control ("CTL") and heterogeneous ("DMF") EDMF schemes.

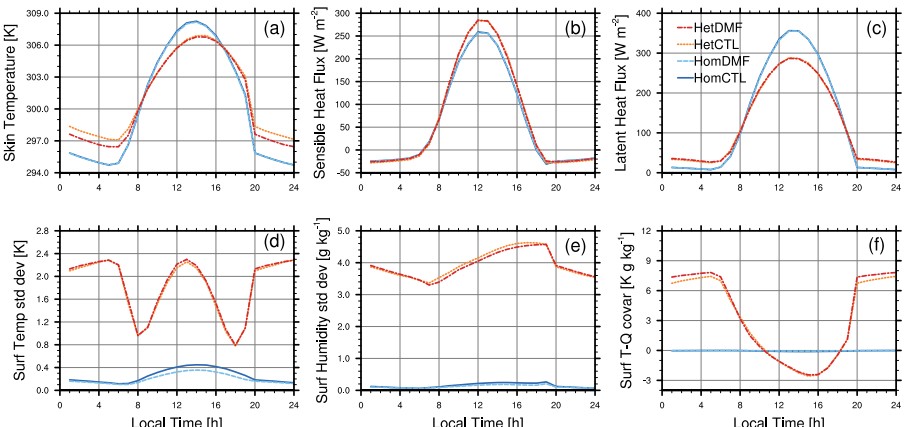

**Figure 2.** Diurnal composite time series of (a) skin temperature, (b) upward sensible and (c) latent heat flux, (d) inter-tile surface temperature and (e) humidity standard deviations, and (f) surface temperature-humidity covariance.

## 4 Results

To provide context for our subsequent analysis of the DMF scheme, in Fig. 2 we compare diurnal composite time series of several surface quantities between the experiments. The grid mean surface skin temperature and surface sensible and latent heat fluxes are shown in Fig. 2a-c. The diurnal cycle of skin temperature is seen to be buffered by the more heterogeneous surface in the Het case, with nocturnal skin temperature increased by 2-3°C, and the daytime peak reduced by 2°C. Nocturnal sensible heat flux is slightly more negative in the Het case, while the daytime peak is somewhat increased. Latent heat flux shows the opposite tendency, with reduced daytime and increased nighttime fluxes in Het. The effect of DMF versus CTL on the mean skin temperature and aggregated fluxes is seen to be small for both cases; aside from nocturnal skin temperature, which is somewhat cooler with DMF, the effect is generally a small fraction of the difference between cases.

The inter-tile variance and covariance of temperature and humidity are shown in Fig. 2d-f. Here we see a dramatic difference between the two cases, with significantly increased variance and covariance in the Het case. The surface temperature variance in Het is minimized in early morning and evening, when the varied diurnal cycles of each surface type bring them into closest agreement (Fig. 3c). The surface humidity standard deviation is likewise much larger in the Het case, with weak diurnal variation in all cases.

Figure 3 provides additional context, with surface properties and fluxes from the HetCTL experiment averaged by surface type. The forest and grassland tiles show similar diurnal variation, though the forest temperature variation is somewhat smaller due to the larger sensible heat fluxes resulting from greater surface roughness. Neither type shows much diurnal variation in humidity, with the forest being somewhat drier and with smaller latent heat fluxes. The lake tile shows qualitatively different behavior, with much smaller diurnal temperature variation that peaks later in the day, and sensible heat fluxes that rise at night, peaking in the early morning.



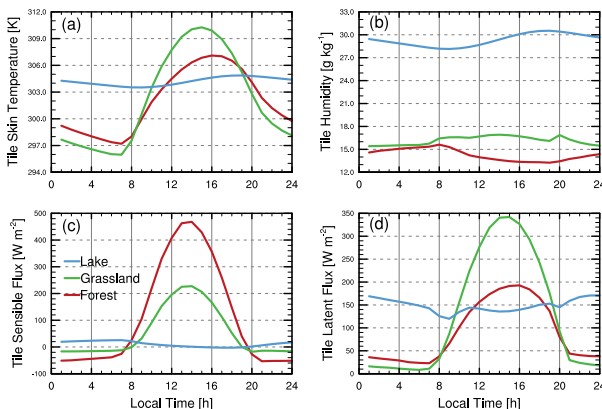

**Figure 3.** Diurnal composite time series from the HetCTL experiment of (a) skin temperature, (b) surface humidity, (c) sensible and (d) latent heat flux, averaged by surface type.

## 4.1 Effect on updraft behavior

A significant difference with DMF relative to the CTL scheme is that updrafts are activated whenever at least one tile has a positive buoyancy flux, even if the grid mean buoyancy flux is negative. Further, the updraft areas are weighted by the relative area of the tile to which they are assigned. The combination of these effects results in more frequent activation of the mass flux scheme, though often with a reduced fractional area relative to the control approach. Figure 4a shows the diurnal composite of the fraction of time that the mass flux is active in the Het case (that is, triggered over at least one tile). With CTL, the mass flux is active continuously between roughly 0800-1600 local time (LT), but is very rarely active at night when the aggregated buoyancy flux becomes negative. In contrast, with DMF the mass flux remains active at night nearly all of the time. The relative source area of the mass flux is reduced, however. Figure 4b shows the surface area fraction associated with active updrafts. For the CTL case (gray bars) this is identical to the active time shown in Fig. 4a. For DMF, the relative contributions from different surface types are shaded as lake (blue), grassland (green), and forest (red). The nocturnal convection, though continuously active, is seen to occur entirely over the lake tile, and thus its fractional area is relatively limited. During the day, although DMF results in convection being always active, on average its properties are drawn from only about 80% of the surface, compared with close to 100% with CTL. This is due largely to reduced convection over the forest and lake tiles at midday, when the surrounding grassland and grid mean air temperature experience a larger diurnal warming (Fig. 3a).

The influence of surface type can be seen in a snapshot of the distributions of updraft properties with height. Instantaneous values from June 3 at 1600 LT in the HetDMF experiment are shown in Fig. 5. Curves indicating the minimum, mean and maximum updraft values of potential temperature (Fig. 5a) and total water specific humidity (Fig. 5b) are color coded by surface type, with lake (blue), forest (red) and grassland (green). At this timestep, the mass flux was active over five grassland tiles, two forest tiles, and the single lake tile. Updrafts originating over the lake are notably cooler and more humid than those over land. Due to the larger number of forest and grassland tiles (as compared to the single lake), as well as their larger sensible

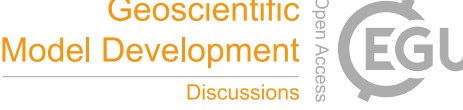

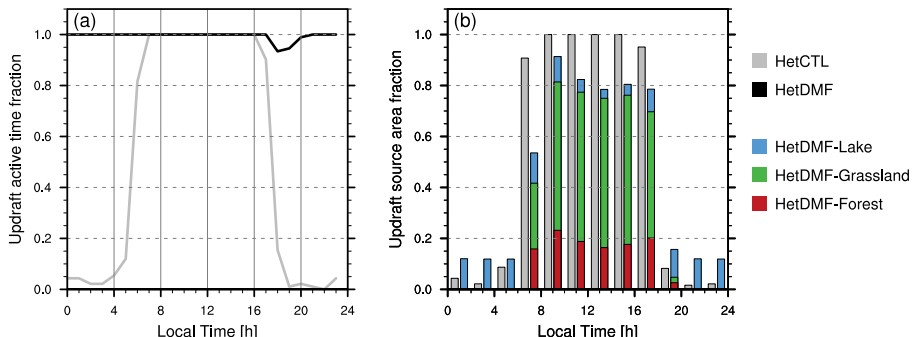

**Figure 4.** Diurnal composites from the Het case of (a) fraction of time with active updrafts for CTL (gray) and DMF (black), and (b) subgrid fractional area acting as updraft source, for CTL (gray) and DMF lake (blue), grassland (green), and forest (red).

heat fluxes, the updrafts over the forest and grassland surface types exhibit a larger spread of temperatures. However, the initial updraft spreads in humidity are comparable, as the large evaporation over the lake tile partially compensates for the absence of

lake heterogeneity.

It is also notable that, although lateral mixing with the environment generally causes the updraft properties to converge with height, updrafts from the various surface types retain distinct temperature distributions up to at least 800 m in this instance, at which point the forest updraft velocities are no longer positive and they detrain. The level at which updraft properties converge may be considered an "updraft blending height," analogous to the blending height at which the atmosphere over

a heterogeneous surface approaches homogeneity (Mahrt, 2000). In this case, the height is a reflection of updraft properties rather than variability at the scale of surface heterogeneity. As such, it would also depend on the specified lateral entrainment rates in the updraft scheme. At the time of this snapshot, mean fractional entrainment rates below 800m were approximately $1.3 \, \text{km}^{-1}$ (but varied stochastically as noted in Section 2.2).

The spread among updrafts is visualized in Fig. 6, which shows the JJA mean from 1200-1600 LT of the inter-updraft

standard deviation for the four primary experiments, conditional on updrafts being present. We may consider two relevant comparisons. First, we note that the DMF scheme produces a larger inter-updraft spread than CTL in both the Hom and the Het configurations; slightly larger in the Hom case, and dramatically so in Het. Second, we may consider the extent to which inter-updraft variability reflects the surface variability in each case. Comparing Hom with Het, we find the CTL scheme produces only a slight increase in inter-updraft temperature variability in the Het case, and a slight reduction in total water variability

despite the much larger surface humidity variance. This is likely a response to the slight increase in daytime grid mean sensible heat flux, and decrease in daytime latent heat flux, since all variation in updraft boundary conditions in CTL is proportional to the surface fluxes, based on Eqs. (4) and (5). By contrast, the DMF scheme produces a consistent increase in updraft variability over the more heterogeneous surface, particularly in humidity, consistent with the difference in surface properties (Fig 2).



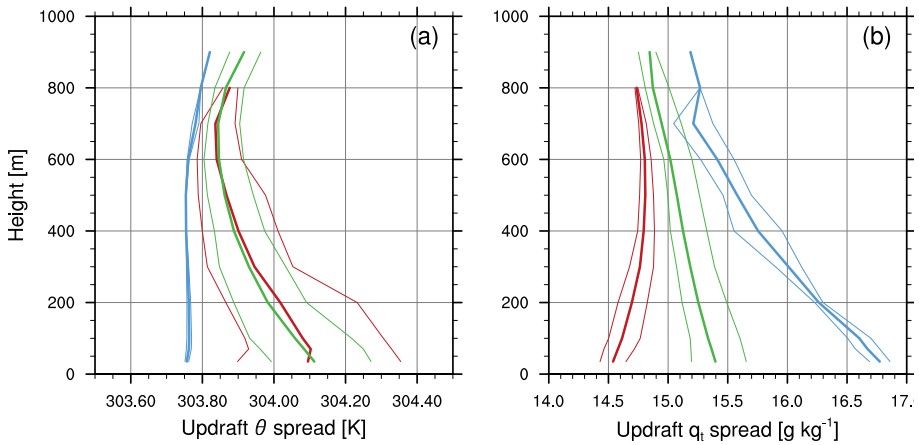

**Figure 5.** Snapshot of the range of (a) potential temperature and (b) total water of updrafts originating over different surface types: lake (blue), forest (red), grassland (green).

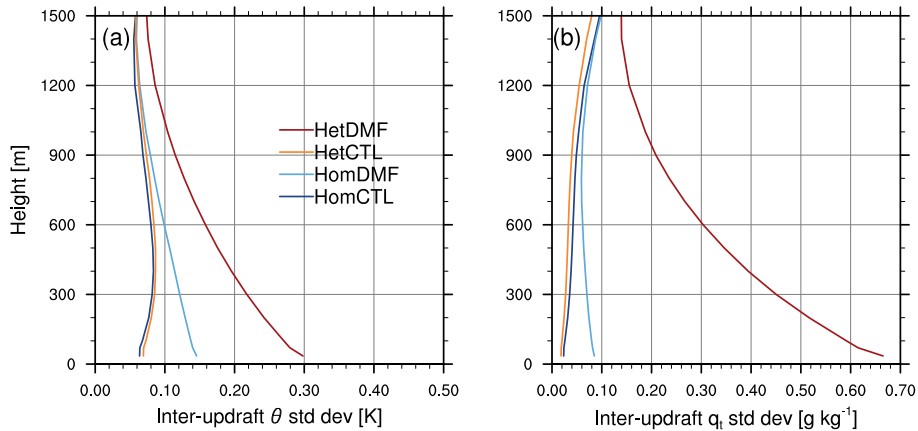

**Figure 6.** The mean inter-updraft standard deviation of (a) potential temperature and (b) total water averaged 1200-1600 LT.

## 4.2 Updraft variance contributions

As described in Section 2.2, a mass flux contribution is included in the estimation of higher order moments, and the more varied updraft thermodynamic properties with DMF might be expected to increase thermodynamic variances, particularly over heterogeneous surfaces. Profiles of the mean afternoon (1200-1600 LT) mass flux contributions to the subgrid variances and covariances are shown in Fig. 7a-d. From left to right, these are the variances of liquid water static energy, $\overline{s_L'^2}$, total water specific humidity, $\overline{q_t'^2}$, the covariance of the two, $\overline{s_L'q_t'}$, and the variance of vertical velocity, $\overline{w'^2}$. In the Hom case, the DMF

approach largely reproduces the MF contributions from CTL, with perhaps a small increase in near-surface $s_L$ variance and $s_L$-$q_t$ covariance. This similarity might be expected given the relatively homogeneous surface. However, in the Het case, DMF





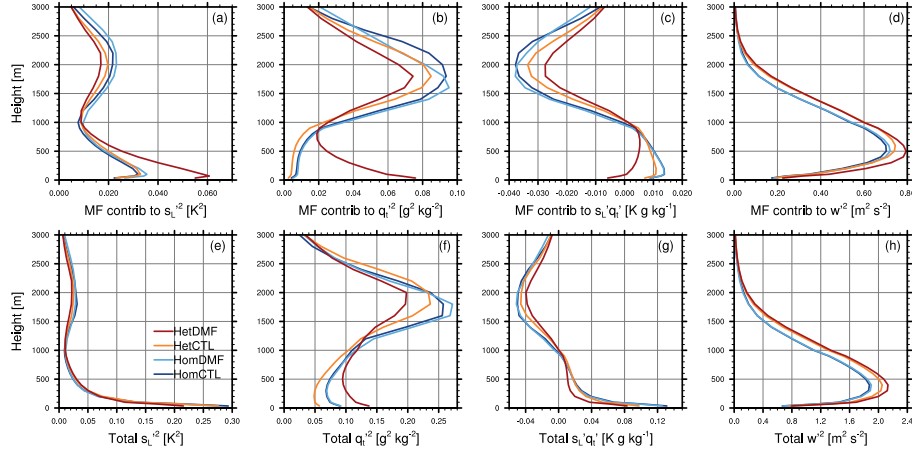

**Figure 7.** Mass flux contributions to the subgrid variances of (a) vertical velocity, (b) liquid water static energy, (c) total water, and (d) the $s_l$-$q_t$ covariance. (e-h) The total subgrid (co)variances. All profiles averaged 1200-1600 LT.

produces a significant increase in all three variance contributions, and the MF contribution to covariance changes from positive to negative. The CTL approach shows much smaller dependence on the surface (Hom versus Het), with almost no change in the contributions to $s_L$ and w variance, and slight decreases in $q_t$ variance and $s_L$-$q_t$ covariance. The reduced contribution to
$q_t$ variance is particularly notable given the much larger surface humidity variance in the Het case.

The mean afternoon profiles of total (co)variances are shown in Fig. 7e-h. Differences between CTL and DMF are generally consistent with the changes in MF contribution evident in Fig. 7a-d. The single exception is the total $s_L$ variance, which is somewhat reduced in HetDMF relative to HetCTL, despite the increased MF contribution.

### 4.3 Impacts on the mean state

In principle, the DMF approach can impact the mean state by altering the updraft vertical fluxes, and by modifying the higher order moments used as inputs to the ADG PDF. This in turn can affect cloud properties, buoyancy flux and the generation of TKE. Profiles of several mean state variables averaged 1200-1600 LT are shown in Figure 8a-d. The liquid water static energy and total water mixing ratio are warmer and drier below 1500 m in the Het cases relative to Hom, but differences between CTL and DMF are quite small, with a mean warming of roughly 0.1 K and drying of 0.1 g kg$^{-1}$ associated with DMF. Similarly,
profiles of TKE are nearly identical in the Hom experiments, though in the Het experiments DMF is associated with a slight increase in TKE from 1000 m to 1500 m. Fig. 8d shows a reduction in cloud fraction and increase in cloud base in both Het experiments, with very small increases in the peak cloud fraction with DMF.

The small changes seen in the mean thermodynamic profiles may result in part from the relaxation tendencies. Although the relaxation timescale is quite long - 48 h above 1 km and nearly 50 d at 100 m height - the mean relaxation tendencies shown in
Fig. 9 are seen to shift so as to reduce the thermodynamic changes associated with DMF.





Diurnal composite time-series are shown in Fig. 10. Like the mean profiles, the boundary layer height (BLH), defined as the height at which the diffusivity profile first decreases to $2\,\mathrm{m^2s^{-1}}$, is seen to depend more strongly on the surface characteristics than on the DMF approach. In all experiments, the depth is seen to rise from 100-300 m at night to a mid-afternoon peak of approximately 1500 m. The depth remains 100-200 m deeper in the Het experiments, both day and night. Differences

between CTL and DMF are small, although there is a slight increase in daytime BLH in the HetDMF case, consistent with the elevated TKE seen in Fig. 8c. It is unclear if this is a consequence of the DMF approach, as the maximum updraft depth varies little between the experiments (Fig. 10c), though differences in updraft thermodynamic properties could potentially impact the generation of TKE. At night, both Hom and Het cases show a modest increase in updraft depth with DMF, but the depth remains relatively shallow, as the updrafts seem unable to penetrate the residual layer aloft even with the lake-influenced lower

boundary conditions.

The low cloud fraction, defined as the maximum cloud fraction below 700 hPa, is shown in Fig. 10b. There is generally less cloud fraction in the Het experiments compared with Hom, particularly at night. The cloud fraction is slightly larger in the HetDMF experiment. Figure 10d shows the updraft cloud base, averaged conditionally on cloudy updrafts being present. The daytime cloud base is slightly higher in the Het case, consistent with Fig. 8d, but little difference is seen between CTL

and DMF during the day. At night however, both HomCTL and HetCTL include periods when there are no cloudy updrafts, whereas this occurs less frequently in HomDMF, and in HetDMF at least one updraft reaches its condensation level for each hour of the composite.

### 4.4   Parameter sensitivities

In this section we examine changes in updraft spread as several key parameters are varied. This is intended to highlight the

influence of uncertain parameters within the scheme, as well as potential sensitivities to EDMF parameters that may differ across models. The $\beta$ parameter, determining the proportionality of tile-scale variability between the surface and atmosphere, was set to $0.25$ in our primary experiments but is a significant unknown. Figure 11a,b shows the inter-updraft standard deviation for the HetDMF case with $\beta$ values between 0 and $0.75$. For both potential temperature and total water, the near-surface standard deviation is seen to increase monotonically as $\beta$ is increased. The enhanced updraft variability due to the $\beta$ term decays with

height, but remains visibly increased through at least 1500 m in all cases except $\beta = 0.75$.

The dependence on updraft lateral entrainment rate is shown in Fig. 11c,d. The entrainment rate is a common tuning parameter in mass flux schemes and would be expected to modulate the impact of the DMF approach, as the lateral entrainment process acts to bring each updraft's thermodynamic properties closer to the grid mean, thus reducing the inter-updraft spread. To examine this sensitivity, we vary an entrainment scaling factor $\epsilon_0$ from $0.15$ to $0.35$, from its default value of $0.25$. The

near-surface sensitivity to $\epsilon_0$ is much smaller than to $\beta$, but away from the surface a clear shift to smaller standard deviations is visible as $\epsilon_0$ is increased.

Finally, we examine the dependence on the number of updrafts, $N$, in Fig. 11e,f. Like the entrainment rate, the specified number of updrafts can vary depending on the EDMF implementation (Suselj et al., 2021; Witte et al., 2022; Chinita et al., 2023). If the updraft number is similar to the number of surface tiles (10 in these experiments), it becomes more difficult to



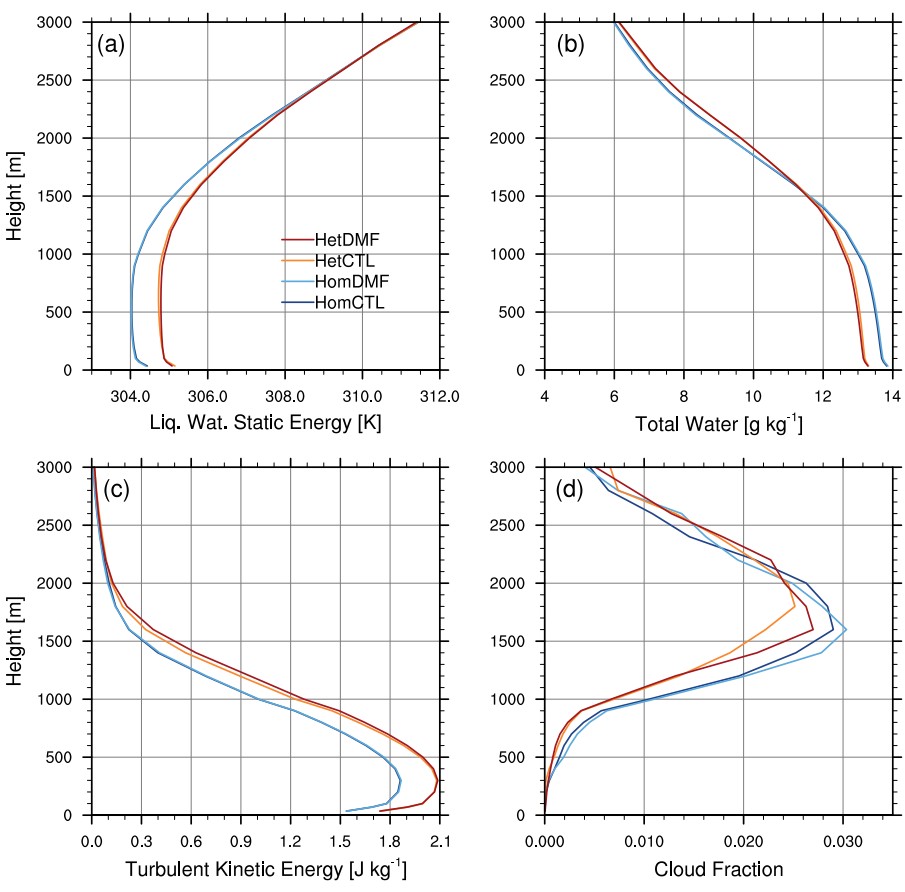

**Figure 8.** Profiles averaged 1200-1600 LT of (a) liquid water static energy, (b) total water mixing ratio, (c) turbulent kinetic energy, and (d) cloud fraction.

represent the intra-tile variability. In the limit of a single updraft per tile, the intra-tile variance is unrepresented, and updraft spread is primarily due to inter-tile variability. Decreasing $N$ from 30 to 10, we find a relatively weak dependence. The standard deviation in this case varies non-monotonically with $N$, with a somewhat larger variation near the surface that decreases with height. This suggests that the intra-tile contribution to the inter-updraft variance is relatively small.

## 5   Conclusions

This study examined a new method to represent the effects of surface heterogeneity on shallow updrafts in a multiple plume EDMF parameterization. The approach involves distributing EDMF updrafts across subgrid surface elements to allow propagation of surface characteristics into the boundary layer. Updraft lower boundary conditions are drawn from assumed joint normal distributions for vertical velocity and thermodynamic variables defined over each individual surface tile based on tile-level surface fluxes and inter-tile surface anomalies relative to the grid mean.



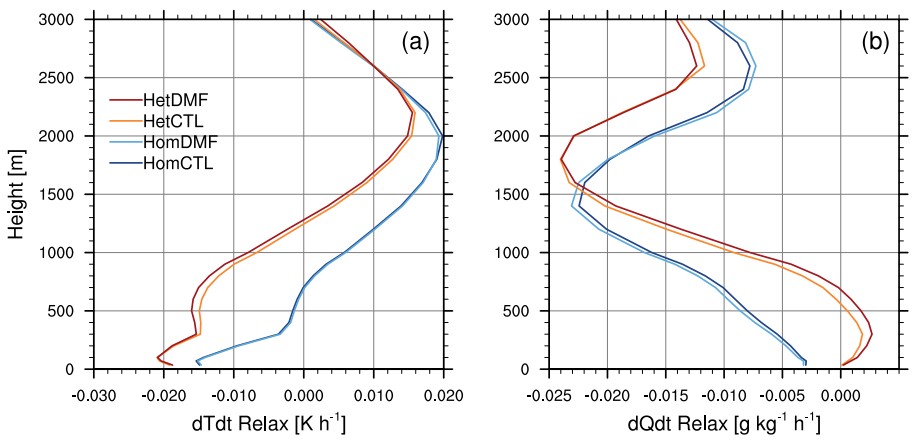

**Figure 9.** Relaxation tendencies of (a) temperature and (b) humidity averaged 1200-1600 LT.

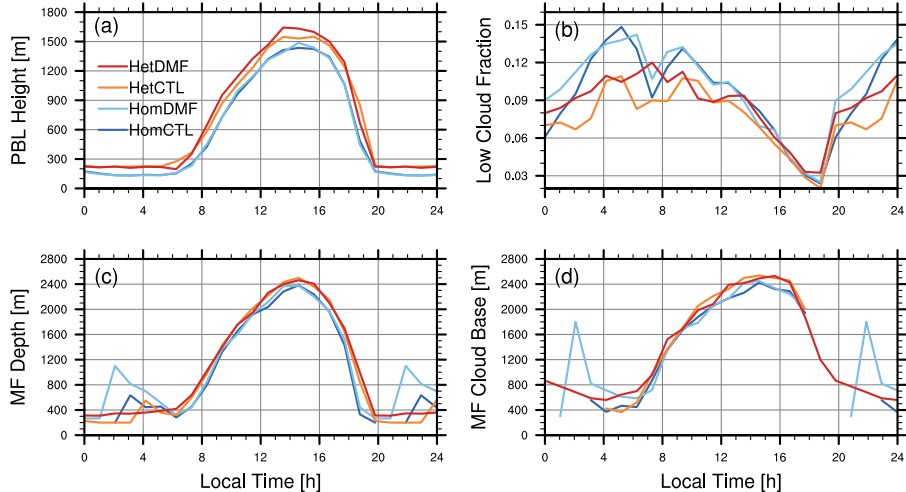

**Figure 10.** Diurnal composite time series of (a) boundary layer height, (b) low cloud fraction, (c) updraft depth, and (d) updraft cloud base. Updraft properties are averaged conditional on the presence of updrafts or cloudy updrafts.

This Distributed Mass Flux (DMF) approach was studied in a set of experiments with the NASA GEOS single column model over the ARM SGP site, with both a realistic surface and an enhanced heterogeneity case that included forest tiles and a lake. The approach was found to modify updraft properties as expected, with larger inter-updraft variation over the more heterogeneous surface. Groups of updrafts assigned to different surface types were shown to inhabit distinct thermodynamic distributions; for example, updrafts originating over a lake tile at mid-day being more humid and cooler. The mass flux contribu-

tions to estimates of subgrid variances - vertical velocity, liquid water static energy and total water - also co-varied with surface conditions in a physically intuitive way. The DMF approach produced larger near-surface variances over the heterogeneous surface for all three variables. By contrast, the control approach showed only a weak dependence on surface heterogeneity.



**Figure 11.** Inter-updraft standard deviation of potential temperature (left) and total water (right) for the Het case, as a function of the $\beta$ parameter (a,b), the updraft lateral entrainment rate (c,d), and the number of updrafts (e,f).





Parameter sensitivities were also examined. The spread of updraft thermodynamic properties was found to be relatively insensitive to both the specified number of updrafts and the updraft lateral entrainment rate. However, the spread was seen to depend strongly on the $\beta$ parameter, which determines the proportionality of tile-scale atmospheric variability to the surface tile deviations from the mean. Larger values of $\beta$ were associated with greater inter-updraft variability. Though it is a fixed constant in our experiments, in principle $\beta$ could be made a function of the stability, wind speed, spatial scale of heterogeneity, and other factors that determine the coupling between surface and near-surface atmospheric heterogeneity.

A potential limitation of the DMF approach is the need for a sufficient number of updrafts to sample both the inter-tile and sub-tile variances. If the number of updrafts is comparable to or smaller than the number of surface tiles, many tiles will be represented by a single updraft. Within a strongly heterogeneous grid box in which inter-tile variability exceeds the estimated sub-tile variability, thermodynamic properties may still vary appropriately across the updraft ensemble due to the inter-tile $\beta$ term. Indeed, our results indicated that using 10 updrafts (rather than 30) in the Het case made little difference to the inter-updraft thermodynamic spread. However, over a homogeneous surface where inter-tile variability is negligible, assigning a single updraft per tile could result in almost uniform updraft properties. This could be addressed by increasing the number of updrafts, though of course with additional computational cost. Another possibility would be to require a minimum number of updrafts per tile in order to represent the sub-tile variability, with such updraft groups distributed over the most buoyant tiles. One can imagine more sophisticated strategies, in which updrafts are apportioned among buoyant tiles in order to optimally represent both the sub-tile and inter-tile variances.

In our implementation, the updraft fractional area is made proportional to the surface source area. That is, if surface buoyancy flux is positive over only half the surface area, the potential updraft area will be halved. If paired with a distribution strategy that limited updrafts to a subset of the buoyant surface area, this approach could artificially restrict the updraft area fraction and with it any tendencies due to the mass flux. This could be avoided by re-scaling the updraft area to match the total buoyant surface area, though this would technically be inconsistent with the assumed tile-level normal distributions.

Despite the modified updraft properties, impacts on the mean state variables were found to be quite modest. The afternoon mean profiles of temperature and total water were largely unaffected by the DMF approach, while cloud fraction increased slightly. It is possible this results from our use of relaxation tendencies to constrain the SCM experiments; although the tendencies in the lower ABL are quite small, they do change so as to reduce differences between experiments. Another possibility is that, although updraft variability is enhanced with DMF, the mean updraft fluxes and resulting tendencies are less affected due to an approximate balance between positive and negative updraft anomalies.

A more significant mean state effect might be obtained with further modifications to the EDMF scheme. Many studies have pointed to secondary mesoscale circulations as an important mediator of the effects of surface heterogeneity (Simon et al., 2021). These can transport moist air from relatively humid regions to areas where strong sensible heat flux drives ascent and additional cloud formation. It may be possible to modify the updraft model within an EDMF framework so that the dynamics of one or more plumes were appropriate for mesoscale ascent. When conditions warrant, such a "mesoscale plume" could be triggered over the most buoyant subgrid region, with lower boundary conditions reflecting a mesoscale inflow area. We leave exploration of this idea to future work.



*Code and data availability.* The baseline GEOS model code is available at https://zenodo.org/records/10413199 (GMAO, 2023). The code modifications for the heterogeneity scheme and single column model output used in this paper are archived on Zenodo at https://zenodo.org/records/10414629

(Arnold, 2023).

*Competing interests.* The author declares that no competing interests are present.

*Acknowledgements.* The author thanks NASA program manager David Considine for supporting this project as part of the Coupling of Land and Atmosphere Subgrid Parameterizations (CLASP) Climate Process Team (CPT). This work relied on computing resources provided by the NASA High-End Computing (HEC) Program through the NASA Center for Climate Simulation (NCCS) at Goddard Space Flight Center.





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
