# Peer review of "Representing effects of surface heterogeneity in a multi-plume eddy diffusivity mass flux boundary layer parameterization"

_Geoscientific Model Development, 2023_

## Referee Comment (RC2)

**Review of "Representing effects of surface heterogeneity in a multi-plume eddy diffusivity mass flux boundary layer parameterization" by Nathan Arnold**

**Summary**

The author presents a modeling approach that allows communication between subgrid surface heterogeneity and the overlying atmosphere by accounting for the updrafts assigned to individual buoyant surface tiles. This approach is incorporated in a multiple plume EDMF boundary layer scheme and configured in the single column mode of the GEOS model to assess the impact on the boundary layer states and shallow clouds. Sensitivities of major parameters in the proposed approach are also examined to understand the uncertainty that may be introduced. Results show that the new approach can more effectively reflect the surface heterogeneity compared to the original treatment in GEOS through the inter-updraft variation of thermodynamic quantities, though it has a pretty modest impact on the mean states and cloud properties in the boundary layer. Limitations and pathways toward future implementation in ESMs are also discussed. This work is useful in the land and atmosphere modeling communities and within the scope of GMD. The manuscript is well written and organized. I recommend publication provided that the below minor comments are addressed.

**Minor comments**

1. L118: "singly to the most buoyant tiles in descending order" It is a bit confusing. Do you mean the remainder of the updrafts are assigned to those least buoyant tiles?

2. L164: "We use 137 levels" Is 137 a typical vertical level number for 0.5-deg resolution GEOS? It seems too many for a coarse-resolution Earth system model. Does the author think the vertical resolution will influence the DMF performance (or the propagation of the surface heterogeneity upward the atmospheric boundary layer)?

3. L176: "heterogeneous"->"modified" to distinguish from the heterogeneous surface case just being mentioned before.

4. Figure 5: (1) please add a legend for Fig. 5 (2) using an error bar to denote the data range may be more appropriate here

5. Figure 8: Could the author elaborate more about the cloud fraction changes due to the applied DMF approach? Is it a robust signal related to the heterogeneous treatment of the updrafts? Or is it arising from a couple of profiles with large cloud fractions? Considering the simulation period is only three months with a mean cloud fraction of ~2%, a small number of overcast profiles might determine the statistics. If it shows consistently increased cloud fraction in DMF, could the author discuss more about the underlying mechanisms? The author stated in L255 that "the DMF approach can impact the mean state by altering the updraft vertical fluxes, and by modifying the higher order moments used as inputs to the ADG

PDF. This in turn can affect cloud properties, buoyancy flux and the generation of TKE.", which apparently is not demonstrated in Fig. 8. It might be helpful to add profiles of vertical fluxes, higher order moments, buoyancy flux, etc., for clarifying the impact of DMS in model simulations.

---

## Author Comment (AC1)

NASA Goddard Space Flight Center
Global Modeling and Assimilation Office
8800 Greenbelt Rd. Greenbelt, MD 20771
Tel: (301) 614-5651, nathan.arnold@nasa.gov

Christopher Horvat
Topic Editor

Re: Manuscript GMD-2023-245

Dear Editor,

We found the reviewer comments constructive and useful, and we have carefully addressed all suggestions in our revision. The most significant changes include:

- A new figure with histograms of low cloud fraction, to provide context for the apparent changes in low cloud;

- The inclusion of profiles from the $\beta = 0.5$ experiment in Figs. 7 and 8, to show how the DMF impacts change with a larger value of $\beta$;

- A new table indicating the fractional areas of surface tiles for each vegetation type.

A point-by-point response to all reviewer comments can be found below. We feel the manuscript has benefited significantly from this revision, thank the reviewers for their comments, and thank you again for your consideration.

Sincerely,

Nathan Arnold

Reviewer comments in blue

**Reviewer #1**

The manuscript presents and analyzes a new approach to allow subgrid scale heterogeneity in land surface to influence updraft thermodynamic properties in an EDMF scheme. A series of single column model experiments were conducted with conditions from the ARM SGP site in the summer of 2017, and the results showed an increase in the near-surface subgrid scale variances, and a relatively small impact on the mean state. Sensitivities to parameters were also conducted, and a discussion of the further development of the scheme to increase the impact on the mean state was included in the discussion.
The modification to the EDMF scheme has the potential to allow for the impacts of surface heterogeneity to have an impact at higher levels in the atmosphere at relatively small computational cost, and the continuation of the development of DMF as suggested in the discussion is warranted. The contribution to modeling of turbulent processes in a GCM of the DMF scheme makes this manuscript of interest to atmospheric modelers. I recommend publication of the manuscript with the minor revisions suggested here.

Thank you for the summary and constructive comments.

Line 35 - Perhaps a little too simple - suggest "...by modifying the updraft properties of individual plumes in the context of an EDMF scheme" or something like that.

I have changed the sentence:
"In this study, we  incorporate heterogeneity into an ABL scheme by modifying the lower boundary conditions of individual updrafts in the context of an Eddy Diffusivity Mass Flux (EDMF) parameterization."

Sections 2.2, 2.3 - Although the descriptions of EDMF and SHOC are quite elegant and clear, I am going to suggest that the level of detail included in these sections is not needed as part of this manuscript. What is needed are the deviations in GEOS from the more standard implementations in other model, and enough detail to understand the additions to

I agree that some of the detail in these sections was unnecessary to understand the new aspects of the scheme, and some of the less relevant aspects have now been removed. I did retain equations (3) and (4) in which lower boundary conditions are specified, as these are directly modified in the DMF approach. I also kept the EDMF decomposition (Eq. 1), and the plume equation (Eq. 2), as I think these provide useful context.

> Line 168: Please provide the fractional coverage for the different tiles in the "Het" case.

A table has been added with the fractional area and vegetation type for all tiles in the Het case:

| Fractional Area | Surface Vegetation Type |
| --- | --- |
| .121 | Broadleaf Deciduous Temperate Forest |
| .072 | Broadleaf Deciduous Temperate Forest |
| .070 | Broadleaf Deciduous Temperate Forest |
| .021 | Broadleaf Deciduous Temperate Forest |
| .338 | Grassland |
| .046 | Grassland |
| .106 | Grassland |
| .074 | Grassland |
| .031 | Grassland |
| .121 | Lake |

> Line 170: A sentence or two addressing the use of the "observed" Q1 and Q2 with the "Het" surface case is warranted. Are they still the appropriate forcing terms?

Thank you, this is a good point. The large-scale advective forcing tendencies were derived with a "realistic" case in mind (i.e., Hom), and may be inappropriate for the altered surface conditions in Het. Even in the Hom case, the land properties will differ from observations and potentially be inconsistent with the atmospheric forcing, because the surface tiles are allowed to freely evolve based on an imperfect model. Similarly, the analyzed temperature and humidity profiles used in the relaxation terms may be out of equilibrium with the surface.

We now comment on these issues on line 175:

Being derived from observations, the VARANAL forcing and analyzed profiles used for relaxation should be appropriate for a "realistic" case such as Hom, but may be inconsistent with the altered surface conditions in the Het case. Although synoptic scale advective tendencies would be minimally affected by a local forest or lake, one might expect larger changes in the near-surface temperature and humidity profiles. Given that the surface is freely evolving in all experiments, even the surface in the Hom case could be somewhat inconsistent with the VARANAL profiles. Ultimately, we believe this merits some caution when interpreting differences between Hom and Het. However, given that the same forcing and surface conditions are used in both the CTL and DMF experiments, this should not impact our conclusions regarding the impacts of DMF.

> Line 182: Please comment on why the Het case (with decidious trees and a lake) has larger daytime sensible heat and a smaller daytime latent heat flux than the grassland Hom case. The higher skin temperature for the Hom case and perhaps an intuitive expectation could suggest the opposite. For example, does the forest tile have a smaller vegetation cover area than the grassland tile? Mention is made one line 193 of greater surface roughness, but this would increase both sensible and latent heat fluxes.

The sensible flux is larger in the Het case due to the increase in surface roughness associated with the forest tiles, which is enough to offset the lower surface temperature. The reduced latent flux stems from a lower soil moisture in the Het case, which we now discuss on line 199:

"The forest and grassland tiles show similar diurnal variation, though the forest temperature variation is somewhat smaller due to the larger sensible heat fluxes resulting from greater surface roughness. Neither type shows much diurnal variation in humidity, with the forest being somewhat drier and with smaller latent heat fluxes.  This difference stems from a roughly 20% lower soil moisture on the forest tiles, which is present in the initial conditions and persists through the experiment. The near-surface atmospheric humidity decreases by a smaller amount, limited in part by the relaxation tendency. The result is a smaller land-atmosphere humidity difference and reduced latent heat flux. "

Line 205: There are several fractions being discussed here and above, and the relationships among them warrants some clarification: Fractional coverage of each tile type; fractional area of updrafts within each tile; fractional time the (any?) updrafts are active.

This paragraph has been changed to clarify the ambiguous fraction references:

"A significant difference with DMF relative to the CTL scheme is that updrafts are activated whenever at least one tile has a positive surface buoyancy flux, even if the grid mean buoyancy flux is negative. Further, the updraft areas are weighted by the relative area of the tile to which they are assigned. The combination of these effects results in more frequent activation of the mass flux scheme, though often with a reduced updraft fractional area relative to the control approach. Figure 4a shows the diurnal composite of the fraction of time that the mass flux is active in the Het case (that is, triggered over at least one tile). With CTL, the mass flux is active continuously between roughly 0800-1600 local time (LT), but is very rarely active at night when the aggregated buoyancy flux becomes negative. In contrast, with DMF the mass flux remains active at night nearly all of the time. The relative source  tile area is often reduced, however. Figure 4b shows the  diurnally composited tile area fraction associated with active updrafts. For the CTL case (gray bars) this is identical to the active time shown in Fig. 4a. For DMF, the relative contributions from different surface types are shaded as lake (blue), grassland (green), and forest (red). The nocturnal convection, though continuously active, is seen to occur entirely over the lake tile, and thus  the source tile and updraft fractional areas  are relatively limited."

The higher order moments have no direct impact on microphysics, but can influence the macrophysics. We added the following clarification:

"In principle, the DMF approach can impact the mean state by altering the updraft vertical fluxes, and by modifying the higher order moments used as inputs to the ADG PDF. For example, changes in the variance or skewness of the subgrid total water can change the fractional area and water amount exceeding saturation,

directly modifying the cloud fraction and condensate. This in turn can affect the diagnosed liquid water flux, buoyancy flux and the generation of TKE."

> Line 260 and Figure 8: Are there differences in condensate (cloud water here I assume)? Panels a and b include this information but may mask the role of liquid water. Cloud fraction may not adequately describe the impact of the differences in cloud properties. I recommend adding cloud water, replacing cloud fraction with cloud water, or just reporting on differences.

The cloud fraction profiles in Fig. 8 have been replaced with cloud liquid condensate, and the accompanying discussion has been changed as follows:

Fig. 8d shows a reduction in cloud  liquid condensate and an increase in cloud base  height in the Het experiments. Perhaps the largest mean state difference with DMF is a 10-20% reduction in the peak cloud condensate relative to CTL. This is accompanied by very small increases in  peak cloud fraction, though with small decreases in cloud fraction at other heights (not shown).

However, note that a comment from Reviewer 2 led to significant caveats on the cloud changes (see final comment below).

> Line 265: Have you tried an experiment without the relaxation? The need for the relaxation terms may not manifest at the start of the simulation, so an experiment without the relaxation may shed light on the impact of the relaxation.

I did run the final CTL experiments without relaxation, but not the DMF experiments. At an earlier stage of this research I ran all experiments without the relaxation and focused my analysis on the initial two weeks. However, it became apparent that differences between experiments were dominated by random variability and drift, rather than systematic changes due to DMF. This motivated the use of the relaxation terms and the longer time period (a full JJA).

> Line 290: Can you speculate on the impact of the 0.75 Beta parameter on the conclusions of this study? Would you have seen a bigger impact on the mean flow?

This is a very good question. Rather than speculate, I have added the 0.5 Beta

results for the HET case to the profiles of Figures 7 and 8. There is indeed generally a larger impact. This is now discussed in Section 4.4:

The value of $\beta$ also modulates the impact of DMF on the mean profiles and higher order moments. Variance profiles from the $\beta = 0.5$ experiment are shown in Fig. 7 (red dashed lines). The near-surface impact of DMF is seen to scale roughly in proportion to $\beta$, with the exception of the total $w'^2$, which is almost unchanged. Impacts also begin to emerge in some of the mean profiles in Fig 8 (red dashed lines), with an approximately 0.25 K cooling in the lowest 300 m, and a 15% reduction in the peak TKE. However, the total water profile remains largely unchanged. The condensate profile is shifted downward by 200 m, with notably more cloud below 1 km. This increase in near-surface cloud appears to result from the larger thermodynamic variances (Fig. 7a,b) and reduced temperature (Fig. 8a), which occasionally cause a small subgrid fraction to exceed saturation. However, the peak cloud remains approximately the same as with $\beta = 0.25$, and similarly, systematic changes are not obvious in a histogram of low cloud fraction (not shown).

A byproduct of increasing $\beta$ to 0.5 is a reduction in the daytime source area fraction, from approximately 0.8 to 0.65. With larger $\beta$ terms, updrafts over tiles with below average surface temperature become increasingly negatively buoyant and are reassigned to other tiles (see Section 2.4). This has the effect of shifting the initial updraft temperature anomalies further positive, which likely causes the near-surface cooling seen in Fig. 8a. The magnitude of these effects increases further with $\beta = 0.75$. In our view, values of 0.1 to 0.5 represent a reasonable tuning range for $\beta$, but this should be further explored using LES and observations over a wide range of conditions.

[Figure]

**Figure 7.** Mass flux contributions to the subgrid variances of (a) vertical velocity, (b) liquid water static energy, (c) total water, and (d) the $s_l$-$q_t$ covariance. (e-h) The total subgrid (co)variances. All profiles averaged 1200-1600 LT.

[Figure]

**Figure 8.** Profiles averaged 1200-1600 LT of (a) liquid water static energy, (b) total water mixing ratio, (c) turbulent kinetic energy, and (d) cloud fraction.

Line 340 - The experiment design (even the het case) may not include enough sub-grid scale heterogeneity to fully explore the impact of DMF. The lake tile is small and the grassland and forested tiles are relatively similar. Perhaps a coastal grid box or half a box of bare soil tiles would exhibit a larger impact of DMF. Perhaps a more heterogeneous Het case would be worth exploring.

Thank you, this possibility is now noted in the Conclusions:

Another possibility is that, although updraft variability is enhanced with DMF, the mean updraft fluxes and resulting tendencies are less affected due to an approximate balance between positive and negative updraft anomalies. The heterogeneity in our Het case is also rather modest, and impacts from DMF may be more substantial in a coastal gridbox, or one with a larger lake.

> Line 179 - "among" rather than "between" for more than two in the comparison

Thank you, fixed.

**Reviewer #2**

The author presents a modeling approach that allows communication between subgrid surface heterogeneity and the overlying atmosphere by accounting for the updrafts assigned to individual buoyant surface tiles. This approach is incorporated in a multiple plume EDMF boundary layer scheme and configured in the single column mode of the GEOS model to assess the impact on the boundary layer states and shallow clouds. Sensitivities of major parameters in the proposed approach are also examined to understand the uncertainty that may be introduced. Results show that the new approach can more effectively reflect the surface heterogeneity compared to the original treatment in GEOS through the inter-updraft variation of thermodynamic quantities, though it has a pretty modest impact on the mean states and cloud properties in the boundary layer. Limitations and pathways toward future implementation in ESMs are also discussed. This work is useful in the land and atmosphere modeling communities and within the scope of GMD. The manuscript is well written and organized. I recommend publication provided that the below minor comments are addressed.

Thank you for your summary and constructive comments.

1. L118: "singly to the most buoyant tiles in descending order" It is a bit confusing. Do you mean the remainder of the updrafts are assigned to those least buoyant tiles?

I agree this was confusing and have rephrased as:
"The $N$ updrafts are divided evenly across the buoyant tiles, and any remainder  $R$ is distributed across the $R$ most buoyant tiles."

2. L164: "We use 137 levels" Is 137 a typical vertical level number for 0.5-degree solution GEOS? It seems too many for a coarse-resolution Earth system model. Does the author think the vertical resolution will influence the DMF performance (or the propagation of the surface heterogeneity upward the atmospheric boundary layer)?

The parameterizations used here are based on a development model candidate intended to run with either 137 or 181 levels in a dx=12 km NWP configuration, or

91 levels in a dx=50 km seasonal prediction system. In hindsight, using 91 levels in these experiments would have been more consistent with the target horizontal resolution, but we have found in separate tests that the PBL parameterizations are relatively insensitive to 91L vs 137L (for example, in the variance and covariance profiles). It is likely that propagation of surface heterogeneity would be similarly insensitive. I now note on line 162:

"The baseline boundary layer scheme has been found to be insensitive to vertical resolution in similar continental convective regimes. "

3. L176: "heterogeneous"- "modified" to distinguish from the heterogeneous surface case just being mentioned before.

The text was changed as suggested.

4. Figure 5: (1) please add a legend for Fig. 5 (2) using an error bar to denote the data range may be more appropriate here

A legend was added to Fig. 5. I tried using error bars to show the range, but found that the similarity of the forest and grassland temperature profiles resulted in significant overlap of the bars which made the plot difficult to read.

5. Figure 8: Could the author elaborate more about the cloud fraction changes due to the applied DMF approach? Is it a robust signal related to the heterogeneous treatment of the updrafts? Or is it arising from a couple of profiles with large cloud fractions? Considering the simulation period is only three months with a mean cloud fraction of  2%, a small number of overcast profiles might determine the statistics. If it shows consistently increased cloud fraction in DMF, could the author discuss more about the underlying mechanisms? The author stated in L255 that "the DMF approach can impact the mean state by altering the updraft vertical fluxes, and by modifying the higher order moments used as inputs to the ADG PDF. This in turn can affect cloud properties, buoyancy flux and the generation of TKE.", which apparently is not demonstrated in Fig. 8. It might be helpful to add profiles of vertical fluxes, higher order moments, buoyancy flux, etc., for clarifying the impact of DMS in model simulations.

Thank you for raising this question. I examined time-height plots of cloud variables

and found substantial variability in the differences between CTL and DMF, with multi-day periods of both enhanced and reduced cloud amount. This, along with the histograms below, suggests that the cloud differences seen in Fig. 8 result from random variability rather than a systematic change with DMF. I have added a new figure with histograms of low cloud fraction and a new discussion:

Figure 9 provides additional context for the apparent cloud changes. Histograms of afternoon mean (1200-1400 LT) low cloud fraction, defined as the maximum cloud fraction below 700 hPa, are shown for the four primary experiments. The overall distributions are similar, with relatively small differences between the Hom and Het cases and between CTL and DMF. Within each case the differences between CTL and DMF include both increases and decreases with no obvious dependence on fraction, and the differences within each bin are often inconsistent between Hom and Het. For example, decreases are seen with DMF for fractions 0-0.01 and 0.05-0.1 in the HomSrf case, but HetSrf case shows no change in those bins. This suggests that the mean cloud changes in Fig. 8d are not systematic, but rather result from random variation among experiments.

[Figure]

**Figure 9.** Histograms of afternoon mean (1200-1600 LT) low cloud, from the (a) Hom and (b) Het cases.